# Research on Risk Contagion in ESG Industries: An Information Entropy-Based Network Approach

**DOI:** 10.3390/e26030206

**Published:** 2024-02-27

**Authors:** Chenglong Hu, Ranran Guo

**Affiliations:** Department of Statistics and Finance, School of Management, University of Science and Technology of China, Hefei 230026, China; guorr@ustc.edu.cn

**Keywords:** ESG, tail risk network, information entropy, risk contagion

## Abstract

Sustainable development is a practical path to optimize industrial structures and enhance investment efficiency. Investigating risk contagion within ESG industries is a crucial step towards reducing systemic risks and fostering the green evolution of the economy. This research constructs ESG industry indices, taking into account the possibility of extreme tail risks, and employs VaR and CoVaR as measures of tail risk. The TENET network approach is integrated to to capture the structural evolution and direction of information flow among ESG industries, employing information entropy to quantify the topological characteristics of the network model, exploring the risk transmission paths and evolution patterns of ESG industries in an extreme tail risk event. Finally, Mantel tests are conducted to examine the existence of significant risk spillover effects between ESG and traditional industries. The research finds strong correlations among ESG industry indices during stock market crash, Sino–US trade frictions, and the COVID-19 pandemic, with industries such as the COAL, CMP, COM, RT, and RE playing key roles in risk transmission within the network, transmitting risks to other industries. Affected by systemic risk, the information entropy of the TENET network significantly decreases, reducing market information uncertainty and leading market participants to adopt more uniform investment strategies, thus diminishing the diversity of market behaviors. ESG industries show resilience in the face of extreme risks, demonstrating a lack of significant risk contagion with traditional industries.

## 1. Introduction

Systemic risk is an unavoidable risk that has the potential to trigger widespread disruption and turmoil in financial markets, such as the Chinese stock market crash, the Sino–US trade frictions, and the COVID-19 pandemic. These systemic risks spread rapidly and have a broad impact, swiftly cascading from one market to others [1,2], thereby affecting the stability and growth potential of the entire economy. This risk contagion can lead to interruptions in capital liquidity, credit tightening, and a decline in consumer confidence, exacerbating the risk of economic recession [3]. Furthermore, the implications of systemic risk extend beyond the economic and financial domains, potentially having significant effects on social stability and political situations. In a globalized world, where economies and financial markets are highly interdependent, systemic risk events can quickly spread globally [4], affecting international trade, investment flows, and the operations of multinational corporations.

ESG, the investment philosophy and industry evaluation criteria proposed by the United Nations Environment Programme in 2004, focuses on environmental, social, and governance performance. In recent years, there has been a surge in demand for ESG investments due to the favorable risk/return characteristics of ESG assets [5,6]. However, a stable market environment is crucial for collaborative development, and systemic risks could hinder progress towards achieving carbon neutrality and ecological benefits due to risk contagion among ESG industries [7]. Therefore, researching and revealing the risk contagion within ESG industries is vital for achieving carbon neutrality goals, implementing new development concepts, effectively enhancing the efficiency of financial services to the substantial economy, and supporting economic transformation [6].

This research is driven by two main motivations: (1) to investigate whether there is significant risk contagion among ESG industries during systemic events and how the contagion paths evolve; and (2) to explore whether ESG industries exhibit risk interconnections with traditional industries, meaning whether these industries show interdependent risk response patterns in the face of macroeconomic shocks. Through these motivations, this research aims to provide new perspectives and in-depth analysis for understanding the role of ESG industries in systemic risk management. By analyzing the risk contagion among ESG industries and the risk interconnections between ESG and traditional industries, we hope to offer fact-based insights for investors making investment decisions and assist regulatory authorities in identifying industries that may become centers of risk contagion, and implementing effective regulatory measures to prevent and mitigate potential market turmoil [8,9,10,11].

In financial risk management research, Value at Risk (VaR) provides an important tool for assessing and quantifying the level of systemic risk generated by tail events, and these tools have been widely applied in financial risk management [12,13]. VaR primarily focuses on the risk of individual institutions but shows limitations in measuring the risk of the entire financial system. On one hand, it concentrates on a single institution and fails to measure systemic financial risk; on the other hand, it is also inadequate in effectively measuring the directionality of the risk contagion. To address these limitations of VaR, researchers have further proposed Conditional Value at Risk (CoVaR) [14,15], which addresses these shortcomings by considering the impact of financial institutions on the stability of the entire financial system under specific conditions, especially how risk propagates through different levels of the market in specific systemic risk events [16,17].

Scholars’ quantitative methods for systemic risk spillover using VaR and CoVaR mainly fall into two categories. The first category starts from volatility, focusing primarily on volatility measurement and volatility correlation based on GARCH models. For example, Ganguly [18] explored the risk spillover CoVaR among BRICS economies using the GARCH model. Chen [19] confirmed its effectiveness in predicting tail risks in financial markets. With the increasing complexity of financial market structures, more sophisticated GARCH models have been employed by scholars to research risk contagion among multiple market entities. Canh [20] used a multivariate DCC-GARCH model to research the risk spillover among cryptocurrencies, while Tumala [21] utilized the GARCH-MIDAS model to avoid information loss caused by concatenating or aggregating one variable with another, thus providing new perspectives for research in this field. The second category is based on correlation, encompassing linear models based on regression, such as quantile regression [22,23] models and Single Index Models (SIMs) [24], as well as nonlinear models based on Copula functions [25,26]. Regression models enable researchers to quantify the linear risk spillover relationships between financial entities at different probability levels, while Copula functions allow for the combination of marginal distribution models of different financial entities to capture the nonlinear correlation structure of risk spillovers between them.

As the capital market environment becomes increasingly complex and continues to evolve, its large scale and complex topological structure necessitate the use of network models for quantitative analysis [27]. Complex network models have become key tools for analyzing market risk contagion and inter-industry dependencies. By employing traditional Bayesian networks [28,29] and social network theories [9,30] to analyze the relationships and connection between network nodes, an effective analytical framework is provided for understanding and predicting market risks. These network theories not only assist analysts in identifying key influencing factors within the market but also allow for an in-depth analysis of the interactions and dependencies among market participants through the network’s topological structure. Additionally, network entropy, as a quantitative indicator of network complexity and uncertainty, reflects the randomness and unpredictability of the network state, attracting increasing attention from scholars, especially in using network models to identify the laws of entropy transmission and the risk contagion effect between markets [31,32]. However, current research has limitations in exploring the complex structures of network models under systemic risk shocks, particularly in terms of considering the network structure of tail risks.

To address this issue, this research introduces the Tail Event-Driven Network (TENET) framework to deeply analyze the interconnectedness of tail risks within the ESG industries. It identifies industries of significance within the systemic risk network. This framework, proposed by Härdle [33] and Fan [24], has three distinct advantages. Firstly, it extends the analysis of the bidirectional tail dependencies of CoVaR to a very high-dimensional setting, facilitating the exploration of tail risk spillovers among multiple entities. Secondly, this method combines the Single Index Model (SIM) with variable selection techniques based on the Least Absolute Shrinkage and Selection Operator (LASSO), making it more suitable for describing the complexity among financial market entities. Lastly, the TENET framework constructs a dynamic, directed, and weighted risk contagion network, revealing the dynamics of risk contagion throughout the entire network. For example, Foglia and Angelini [34] analyzed the spillover effects and interconnectivity dynamics among Eurozone financial institutions during a crisis and found that the banking sector contributes significantly to systemic risk. Hernandez [35] explored the risk spillover effects between banks in developed and emerging countries in the Americas through network models, revealing that risk spillover and interconnections among banks in emerging countries are significantly lower than those in developed countries. Xu [36] examined the tail risk dependencies among 23 cryptocurrencies and identified systemically important cryptocurrencies, discovering that Bitcoin is the primary systemic risk recipient, while Ethereum is the largest source of risk output. Zhao [37] researched the financial market risk contagion among the Belt and Road countries and found that countries more affected by the European debt crisis, such as Greece and Cyprus, have relatively higher levels of stock market risk spillover.

Information entropy is an effective tool for measuring system uncertainty or the complexity of information flow. In financial risk contagion research, information entropy can help quantify the efficiency and strength of network risk contagion [38]. For instance, transfer entropy [32,39,40,41,42], which has been used to establish a network model, revealed the network structure and potential paths of systemic risk contagion. Ji [43] employed an entropy-based network method, investigated the risk spillover effects among international real estate markets. Kumar [44] analyzed the interrelations among stocks and the topological characteristics of the Indian stock market from multiple entropy perspectives, offering a new angle to understand market dynamics and intrinsic connections. Entropy not only plays a role in analyzing the internal structure of a single market or financial institution but can also serve as an indicator for measuring network topological properties [45], constructing early warning indicators for systemic risk. Furthermore, Zhao [46] demonstrated that the magnitude of information entropy could reflect the relative importance of financial institutions within the entire financial network, identifying their contribution to systemic risk. However, the limited inter-market risk information that can be collected and the high heterogeneity in the actual network’s connection distribution have been proven to potentially lead to an underestimation of systemic risk [47,48]. Combining the TENET analysis method and the application of information entropy can more accurately predict and assess risks in the market, while also quantifying the efficiency and stability of the entire network structure, providing an effective tool for preventing and responding to potential financial crises.

In terms of contributions, this paper provides two key advancements. Firstly, from a theoretical perspective, this study for the first time applies the method of information entropy to measure the topological characteristics of the TENET network model. This novel approach offers a new perspective for understanding the complexity and dynamics of financial networks. Secondly, in practical terms, by focusing on the risk contagion network within the ESG industries, this research adds a new dimension to the field of financial network studies. Through detailed analysis of the risk transmission pathways between ESG industries, it reveals the mechanisms of risk contagion in ESG industries, providing a crucial foundation for developing effective risk management strategies and promoting the sustainable development of financial markets.

The remainder of this study is organized as follows: Section 2 explores the methods for measuring information entropy within the framework of network models. Section 3 includes some empirical findings, statistical analyses, and the results of related tests. In Section 4, we will discuss and summarize the findings and propose several directions for further research.

## 2. Materials and Methods

### 2.1. Network Model

In the research of financial risk networks, VaR is a key measurement tool. The VaR indicates the potential maximum loss that could occur due to changes in market risk factors at a given confidence level. As financial risk networks involve complex interrelations among multiple financial institutions, assets, and markets, VaR can provide a comprehensive and systematic method for estimating the risk of the entire network. Therefore, VaR is widely used to assess the level of risk within the financial system. The VaR for financial institution *i* at time *t*, given a confidence level α, is defined as:(1)$$P(R_{i,t}\leq VaR_{i,t,\alpha })=\alpha \;$$

Given the confidence level α, $R_{i,t}$ is the return level of asset *i* at time *t*. CoVaR is used to measure the systemic risk contribution of an institution under the condition that another institution is in distress at the same confidence level α, which is a further development based on VaR. CoVaR measures the change in the VaR of the financial system when one institution is in distress. If the distress of an institution increases the VaR of the system, it indicates that this institution has systemic importance. Similarly, the smaller the increase in VaR caused by the distress of an institution, the less systemic importance it has. Therefore, CoVaR can effectively measure the systemic risk contribution of individual institutions within the financial system, and CoVaR is defined as follows:(2)$$P(R_{j,t}\leq CoVaR_{j|i,t,\alpha }|{\bar{I}}_{i,t})=\alpha \;$$
where ${\bar{I}}_{i,t}$ represents the information set, which includes events $R_{i,t}=VaR_{i,t,\alpha }$, $M_{t-1}$, and $B_{i,t-1}$; $M_{t-1}$ represents the vector set of macroeconomic variables that may affect the risk of the financial system; $B_{i,t-1}$ represents the vector set of microeconomic variables that may affect the financial institution *i*.

This research adopts a Single Index Model to estimate VaR and CoVaR [24]. This method is widely used in the field of financial risk management, especially in the assessment and management of market risks of investment portfolios. In this model, the returns of assets are considered to have a linear relationship with other market factors [49]:(3)$$R_{i,t}=c_{i}+\gamma_{i}M_{t-1}+\varepsilon_{i,t}$$
(4)$$R_{j,t}=c_{j|i}+\gamma_{j|i}M_{t-1}+\beta_{j|i}R_{i,t}+\varepsilon_{j|i,t}$$
where $c_{i}$ and $c_{j|i}$ represent the constant terms affecting the returns of financial institutions, $\varepsilon_{i,t}$ and $\varepsilon_{j|i,t}$ are independent and satisfy the normal distribution of residual terms, and $\gamma_{i}$, $\gamma_{j|i}$, and $\beta_{j|i}$ are the risk exposure coefficients representing macroeconomic variables and microeconomic variables, respectively.

Before making these estimates, we must establish some key assumptions. These assumptions provide the necessary theoretical basis for our model and help define its scope of application and potential limitations. $\left(A\right)F_{\varepsilon_{i,t}}^{-1}\left(\alpha |M_{t-1}\right)=0\;\mathrm{and}\;\left(B\right)F_{\varepsilon_{j|i,t}}^{-1}\left(\alpha |M_{t-1},R_{i,t}\right)=\;0$ are assumed [33].

Where F(·) is the distribution function of the residuals, after accepting the above two assumptions, we can estimate VaR and CoVaR using a Single Index Model [34,35]:(5)$${V\hat{a}R}_{i,t,\alpha }={\hat{c}}_{i}+{\hat{\gamma }}_{i}M_{t-1}$$
(6)$${Co\hat{V}aR}_{j|i,t,\alpha }={\hat{c}}_{j|i}+{\hat{\gamma }}_{j|i}M_{t-1}+{\hat{\beta }}_{j|i}{V\hat{a}R}_{i,t,\alpha }$$

In the Single Index Model, the VaR estimation focuses on the risk level of a single asset, while the CoVaR estimation considers the conditional risk level of one asset when another asset is in distress. CoVaR can be used to assess interconnectedness and systemic risk within the financial system. The Single Index Model, by incorporating macroeconomic variables and microeconomic variables that affect the returns of financial institutions, can accurately capture the impact of market dynamics and interdependencies between assets on risk estimation.

After estimating the VaR and the CoVaR, in order to accurately understand and forecast the correlations between the returns of financial institutions, they are included in the information set. VaR and CoVaR are powerful tools for assessing and managing risk; however, they often rely on linear relationships between market variables and returns. In the real world, there are numerous factors that affect the returns of financial institutions, and the relationships between these factors are typically much more complex than what linear models can depict. For instance, extreme market behaviors, changes in macroeconomic policies, and a variety of microeconomic factors could influence returns in a nonlinear manner. Therefore, to more accurately capture these dynamic relationships, it is necessary to construct a new nonlinear model [50]:(7)$$R_{j,t}=g\left(\beta_{j|I_{j}}^{T}I_{j,t}\right)+\varepsilon_{i,t}$$
(8)$${\hat{CoVaR}}_{j|{\tilde{I}}_{j},t,\alpha }^{\mathrm{TENET}}=\hat{g}\left({\hat{\beta }}_{j|{\tilde{I}}_{j}}^{T}{\tilde{I}}_{j,t}\right)$$
where g(·) is a link function, $I_{j,t}=\left\{R_{-j,t},M_{t-1},B_{j,t-1}\right\}$ represents the information set, $R_{-j,t}=\left\{R_{1,t},\ldots ,R_{j-1,t},R_{j+1,t},\ldots ,R_{k,t}\right\}$ is a set of returns that does not include $R_{j,t}$, and the risk exposure coefficient is defined as $\beta_{j|I_{j}}^{T}={\left\{\beta_{j|-j},\beta_{j|M},\beta_{j|B_{j}}\right\}}^{T}$. We can use a rolling window estimation method to estimate all parameters within different windows, where ${\tilde{I}}_{j,t}=\left\{{V\hat{a}R}_{-j,t,\alpha },M_{t-1},B_{j,t-1}\right\}$, and ${V\hat{a}R}_{-j,t,\alpha }=\left\{{V\hat{a}R}_{1,t,\alpha },\ldots ,{V\hat{a}R}_{j-1,t,\alpha },{V\hat{a}R}_{j+1,t,\alpha },\ldots ,{V\hat{a}R}_{k,t,\alpha }\right\}$ is a set of returns that does not include ${V\hat{a}R}_{j,t,\alpha }$.
(9)$$\begin{array}{cc} \hat{\beta },\hat{g}\left(\cdot \right)\overset{\mathrm{def}}{=} & \arg\min_{\beta ,g(\cdot )}\frac{1}{n}\sum_{j=1}^{k}\sum_{t=1}^{T}\rho_{\tau }(R_{j,t}-g\left(\beta_{j|I_{j}}^{T}I_{j,t}\right)\\ \; & -g^{\prime }\left(\beta_{j|I_{j}}^{T}I_{j,t}\right)I_{j,t}^{T}\beta_{j|I_{j}})\omega_{j,t}\left(\beta_{j|I_{j}}\right)+\lambda \sum_{j=1}^{k}\left|\beta_{j|I_{j}}\right| \end{array}$$
(10)$$\omega_{j,t}\left(\beta \right)\overset{\mathrm{def}}{=}\frac{K_{h}\left(\beta_{j|I_{j}}^{T}I_{j,t}\right)}{\sum_{t=1}^{T}K_{h}\left(\beta_{j|I_{j}}^{T}I_{j,t}\right)},\;K_{h}\left(\cdot \right)=h^{-1}K\left(\cdot /h\right)$$
where $\rho_{\tau }\left(\cdot \right)$ is a loss function for quantile regression. K(·) is a kernel Gaussian kernel, and *h* is a bandwidth. λ represents a tuning parameter. Given all the information sets, using non-parametric estimation, we can estimate ${Co\hat{V}aR}_{j|{\tilde{I}}_{j},t,\alpha }^{\mathrm{TENET}}$, which includes the impact of financial institutions other than *j* and also incorporates the nonlinear relationship reflected by the link function g(·).

This research takes the partial derivatives as the elements of the connectivity matrix for the network. It is worth noting that in the network analysis of this research, only the partial derivatives of institution *j* with respect to other financial institutions are included, and not those with respect to macroeconomic and microeconomic variables. This is because, in the risk network analysis of this research, the focus is solely on the risk spillover effects among institutions:(11)$${\hat{D}}_{j|{\tilde{I}}_{j}}\overset{\mathrm{def}}{=}\frac{\partial \hat{g}\left({\hat{\beta }}_{j|I_{j}}^{T}I_{j,t}\right)}{\partial I_{j,t}}{|}_{I_{j,t}={\tilde{I}}_{j,t}}={\hat{g}}^{\prime }\left({\hat{\beta }}_{j|{\tilde{I}}_{j}}^{T}{\tilde{I}}_{j,t}\right){\hat{\beta }}_{j|{\tilde{I}}_{j}}$$

This approach is based on three key reasons: (1) Sensitivity to marginal changes: ${\hat{D}}_{j|{\tilde{I}}_{j}}$ captures marginal changes in market value loss, reflecting the sensitivity of systemic risk levels to minor fluctuations in market conditions. This reveals the significant impact that even slight market movements can have on the entire system, underscoring the importance of understanding the nuances of systemic risk sensitivity. (2) Accuracy in assessing risk spillover: ${\hat{D}}_{j|{\tilde{I}}_{j}}$ is directly related to the effects of market condition changes on the interdependencies between financial institutions. This allows researchers and policymakers to more accurately identify and quantify the paths through which risk spreads within the network, leading to a more precise evaluation of the strength of the risk contagion. (3) Effectiveness in systemic risk management: By analyzing the intensity and direction of risk spillover among financial institutions, it becomes possible to better identify institutions of systemic importance and potential points of risk concentration. This information is crucial for developing effective risk mitigation strategies and regulatory policies, thereby enhancing the overall management of systemic risk. Its adjacency matrix is as follows:(12)$${\hat{A}}_{t}=\left[\begin{array}{ccccc} 0 & \left|{\hat{D}}_{1|2}^{t}\right| & \left|{\hat{D}}_{1|3}^{t}\right| & \cdots & \left|{\hat{D}}_{1|k}^{t}\right|\\ \left|{\hat{D}}_{2|1}^{t}\right| & 0 & \left|{\hat{D}}_{2|3}^{t}\right| & \cdots & \left|{\hat{D}}_{2|k}^{t}\right|\\ \left|{\hat{D}}_{3|1}^{t}\right| & \left|{\hat{D}}_{3|2}^{t}\right| & 0 & \cdots & \left|{\hat{D}}_{3|k}^{t}\right|\\ \vdots & \vdots & \vdots & \ddots & \vdots \\ \left|{\hat{D}}_{k|1}^{t}\right| & \left|{\hat{D}}_{k|2}^{t}\right| & \left|{\hat{D}}_{k|3}^{t}\right| & \cdots & 0 \end{array}\right]$$
where $\left|{\hat{D}}_{i|j}^{t}\right|$ represents the absolute value of the local gradient of the CoVaR of the institution *i* with respect to the institution *j* in period *t*, and $\left|{\hat{D}}_{j|i}^{t}\right|$ signifies the influence of the institution *j* on the institution *i* under extreme risk conditions at time *t*, that is, the intensity of conditional risk spillover. In essence, ${\hat{D}}_{i|j}^{t}$ captures how institution *i* affects the tail risk of another institution *j* within the risk diffusion network at time *t*.

### 2.2. Transformation of Adjacency Matrices into Stochastic Matrices

Before applying the concept of Shannon entropy, it is necessary to transform adjacency matrices into stochastic matrices [51].

In the field of network theory research, researchers have proposed various methods for measuring the centrality of nodes. Among these methods, eigenvector centrality [52] particularly takes into account the characteristics of a node’s neighbors. The core idea of eigenvector centrality is that the centrality of a node is determined not only by itself but also by the centrality of its neighboring nodes. More specifically, this research assumes that the centrality of nodes is proportional to the weighted sum of their edge scores, where the weights are determined by the centrality scores of neighboring nodes. Given an adjacency matrix **A**, this assumption leads to a set of homogeneous linear equations for the unknown variable of centrality scores $v=\{v_{1},v_{2},\ldots ,v_{k}\}$:(13)$$\sum_{j}a_{ij}v_{j}=\lambda v_{i},\;1\leq i,j\leq k$$
where λ is a proportionality constant. The centrality scores should be non-negative. Equation (Equation 14) elucidates why the proposed centrality is termed eigenvector centrality. For the pair (λ,v), it corresponds to an eigenvalue–eigenvector pair of the adjacency matrix **A**. Imagine that the adjacency matrix **A** is a non-negative matrix. In such a case, the extension Perron–Frobenius theorem [53] reveals to us that there exists a dominant eigenvalue–eigenvector pair $\left(\lambda_{max},v_{max}\right)$ that satisfies the following condition:(14)$$Av_{max}=\lambda_{max}v_{max}$$
where $\lambda_{max}$ and $v_{max}$ are positive values. Normalize $v_{max}$ so that $\sum_{j}v_{max}\left(j\right)=1$. $v_{max}\left(j\right)$ can be regarded as a relative measure of node *j*’s contribution to the entire network. In the context of global financial networks, the right eigenvector $v_{max}\left(j\right)$ reflects the relative contribution of node *j* in terms of risk contagion. For a given adjacency matrix **A**, ‘Method 2’ constructs the following stochastic matrix $P^{*}=\left(p_{ij}^{*}\right)$:(15)$$p_{ij}^{*}=\frac{a_{ij}v_{max}\left(j\right)}{\lambda_{max}v_{max}\left(i\right)},\;1\leq i,j\leq k$$

The components of row *i* of *A* are weighted by ${\left(v_{max}\left(j\right)\right)}_{1\leq j\leq k}$ and normalized by multiplying them by $\frac{1}{\lambda_{max}v_{max}\left(i\right)}$.

### 2.3. Network Entropy

Information entropy is a measure of the uncertainty or complexity of a system, while tail risk focuses on the losses that financial assets or portfolios may suffer in extreme market conditions. By combining these two concepts, we are able to quantify and analyze risk and interdependencies in the financial markets from a novel perspective. We employ the Shannon entropy. For a given discrete probability distribution $D=\{p_{1},p_{2},\ldots ,p_{k}\}$, the formula is defined as follows:(16)$$H\left(D\right)=-\sum_{i=1}^{k}p_{i}\log p_{i}$$

It is noteworthy that H(D) can be interpreted as the expected value of the random variable $\log(\frac{1}{p_{i}})$. Specifically, in the context of financial network analysis, entropy serves as an indicator of diversification, given that the formula assigns higher entropy to random variables with more uniform distributions.

Given a stochastic matrix **P** that is constructed from a network’s adjacency matrix, the entropy $H_{i}$ for a node *i* is determined by applying the entropy formula to the transition probability distribution represented by the *i*th row of the stochastic matrix **P**:(17)$$H_{i}\left(P\right)=-\sum_{j=1}^{k}p_{ij}\log p_{ij},\;1\leq i\leq k$$

$H_{i}\left(P\right)$ measures the diversity of choices of the node *i*. Then, the network entropy $H_{\mathrm{network}}$ is defined as the weighted sum of the entropies of nodes:(18)$$H_{network}\left(P\right)=\sum_{i=1}^{k}\pi_{i}H_{i}$$
where the weighting vector $\pi =\{\pi_{1},\pi_{2},\ldots ,\pi_{k}\}$ is the unique invariant distribution of the corresponding stochastic matrix **P**.

### 2.4. Network Measurement Indicator

TENET indicator [54] measures the interdependence between different institutions and potential risk transmission paths, which helps us to depict the overall health of the financial markets. By monitoring changes in risk-sensitive indicators, regulatory agencies and market participants can promptly identify and respond to potential risk concentration areas, thereby taking preventive measures to reduce the likelihood of systemic collapse.

**Total in degree.** The total in degree represents the extent to which a node is susceptible to the influence of other nodes [55]. A higher total in-degree means that more nodes have a direct impact on it, indicating that this node is more influenced within the network. For a node *j*, its total in-degree at time *t* is defined as follows:
(19)$$TC_{j,t}^{in}=\sum_{i=1}^{k}\left|{\hat{D}}_{j|i}^{t}\right|$$**Total out degree.** The total out degree represents the extent of its influence on other nodes [55]. A higher total out-degree means that the node has a direct impact on a larger number of other nodes within the network, indicating that it has greater influence or contagion capability in the network. For a node *j*, its total out-degree at time *t* is defined as follows:
(20)$$TC_{j,t}^{out}=\sum_{i=1}^{k}\left|{\hat{D}}_{i|j}^{t}\right|$$**Relative influence.** We calculate the relative influence (RI) as the ratio between the difference and the sum of out-tail interconnectedness and in-tail interconnectedness [34]. This indicator enables capturing the sector’s relative impact and magnitude of risk spillover onto other sectors. A positive value signifies that the sector generates more systemic risk than it receives, while a negative value indicates that the sector receives more systemic risk than it generates:
(21)$$RI_{j,t}^{sector}=\frac{TC_{j,t}^{out}-TC_{j,t}^{in}}{TC_{j,t}^{out}+TC_{j,t}^{in}}$$**Centrality of contagion.** The degree to which a node is central in the network indicates the distance of the node from other parts of the network in terms of contagion distance. More central nodes have higher centrality values and are good propagators of shocks. Referring to Abduraimova [56], the centrality of contagion for node *j* at a given time *t* is defined as follows:
(22)$$CONC_{j}^{t}=\frac{1}{\sqrt{{\left(u_{j}^{t}\right)}^{2}+{\left(\sigma_{j}^{t}\right)}^{2}}}$$
(23)$$u_{j}^{t}=\frac{\sum_{i=1}^{k}\left(\left|{\hat{D}}_{j|i}^{t}\right|+\left|{\hat{D}}_{i|j}^{t}\right|\right)}{k-1},\;\sigma_{j}^{t}=\sqrt{\frac{\sum_{i=1}^{k}{\left(\left|{\hat{D}}_{j|i}^{t}\right|+\left|{\hat{D}}_{i|j}^{t}\right|-u_{j}^{t}\right)}^{2}}{k-2}}$$

### 2.5. Mantel Test

The Mantel test [57] has been widely used to assess the null hypothesis of no spatial autocorrelation. Unlike traditional methods of correlation testing that only examine the relationship between two variables, the Mantel test has the unique advantage of assessing the correlation between two matrices **B** and **C**. Its null hypothesis posits that there is no association between the distances among points in one distance matrix and those in a second distance matrix. The essence of the test is to compare whether the correlation among variables within the same group is stronger than between variables of different groups:(24)$$B=\left[\begin{array}{cccc} 0 & b_{12} & \cdots & b_{1k}\\ b_{21} & 0 & \cdots & b_{2k}\\ \vdots & \vdots & \ddots & \vdots \\ b_{k1} & b_{k2} & \cdots & 0 \end{array}\right],\;C=\left[\begin{array}{cccc} 0 & c_{12} & \cdots & c_{1k}\\ c_{21} & 0 & \cdots & c_{2k}\\ \vdots & \vdots & \ddots & \vdots \\ c_{k1} & c_{k2} & \cdots & 0 \end{array}\right]$$

The calculation formula for the Mantel test is as follows:(25)$$M=\frac{\sum_{i=1}^{k}\sum_{j=1}^{k}\left(b_{ij}-\bar{b}\right)\left(c_{ij}-\bar{c}\right)}{\sqrt{\sum_{i=1}^{k}\sum_{j=1}^{k}{\left(b_{ij}-\bar{b}\right)}^{2}}\sqrt{\sum_{i=1}^{k}\sum_{j=1}^{k}{\left(c_{ij}-\bar{c}\right)}^{2}}}$$
(26)$$\bar{b}=\frac{2}{k(k-1)}\sum_{i=1}^{k}\sum_{j=1}^{k}b_{ij},\;\bar{c}=\frac{2}{k(k-1)}\sum_{i=1}^{k}\sum_{j=1}^{k}c_{ij}$$
where $\bar{b}$ and $\bar{c}$ represents the average Euclidean distance. The null hypothesis of this statistic asserts that there is no correlation between the two distance matrices; that is, the degree of similarity of the samples in variable *b* is unrelated to their degree of similarity in variable *c*. When the data exhibit a certain degree of autocorrelation, implying that there is a correlation between matrix **B** and matrix **C**, the null hypothesis is rejected. The significance is evaluated through permutation testing, which involves permuting the corresponding rows and columns of matrix **B** 999 times while keeping matrix **C** constant. The statistic and its significance are calculated for each permutation.

## 3. Results

### 3.1. Data

In this research, 292 outstanding constituent stocks of the China ESG 300 Index are selected as the research subjects. These stocks not only represent companies with excellent Environmental, Social, and Governance (ESG) performance in the Chinese market but also cover a diversified range of industries, demonstrating the profound potential of the Chinese market in sustainable investment. According to the Shenwan first-level industry classification, these 300 constituent stocks are further divided into three major sectors, Consumption, Cycles, and Technology, and subdivided into 28 industries. The specific results are shown in Table 1.

We have selected a range of macroeconomic and microeconomic variables [58] to gain insights into their respective impacts on economic and financial analysis. For macroeconomic variables, we focus on the Consumer Price Index (CPI), which serves as a primary measure of inflation. We also consider the growth rate of the money supply (M0), which is the most liquid measure of the money supply, including coins and notes in circulation and other assets that are easily convertible into cash. Interest rates (R) are another key macroeconomic variable we analyze, as they significantly influence borrowing costs, consumer spending, and overall economic activity. Lastly, we examine the business index of macro-economic (MEI), which provides a holistic view of the economic climate and overall economic health.

On the microeconomic side, this research includes several crucial financial metrics that reflect the health and performance of individual industries. The Price-to-Earnings (P/E) ratio provides insight into the market’s valuation of a company’s earnings. The Price-to-Book (P/B) ratio is also considered, offering an understanding of the market’s valuation relative to a company’s book value. Lastly, the return rates of the Shenwan first-level industry indices are considered, which play an important role in capturing the performance of various sectors within the Chinese economy. These microeconomic variables are critical for investors and analysts to assess the attractiveness of different industries and play a substantial role in portfolio management and strategic investment planning.

Each industry index is calculated using a market capitalization weighting method, which calculates the index according to the proportion of each constituent stock’s market value to the total market value of the industry. The specific calculation formula is as follows:(27)$$Index_{j,t}=Index_{j,1}\times \sum_{i=1}^{N}w_{i}^{j}\frac{P_{i,t}^{j}}{P_{i,1}^{j}}$$
where $Index_{j,t}$ represents the index of industry *j* at time *t*, $Index_{j,1}$ is the base point number of industry *j* at time 1, which is assumed to be 100. $w_{i}^{j}$ represents the market value proportion of the *i*-th stock belonging to industry *j*, and $P_{i,t}^{j}$ represents the closing price of the *i*-th stock in industry *j* at time *t*. The selected research period is from 4 January 2011, to 28 February 2023, with 4 January 2011 being the base period for the index. Stock data are sourced from the Wind database, and logarithmic returns are used when calculating returns. Figure 1 displays the time series data of index returns for the three major ESG sectors, with red representing Consumption, blue representing Cycles, and green representing Technology.

From Figure 1, it is apparent that there was a significant increase in return volatility during certain key periods, such as the stock market crash (2015–2016), Sino–US trade frictions (2018–2019), and the COVID-19 pandemic (2020–2021). These periods may have been marked by severe market fluctuations due to increased market uncertainty, investor panic, liquidity tightening, or changes in macroeconomic policy.

Specifically, during the stock market crash of 2015–2016, concerns over a slowing economy might have eroded market confidence, leading to heightened volatility in industry returns. During the Sino–US trade frictions of 2018–2019, policy uncertainty and tariff threats could have led to shifts in global trade flows and increased market volatility. In the period of the COVID-19 pandemic outbreak from 2020 to 2021, the global impact of the pandemic and widespread restrictions on economic activity further exacerbated market fluctuations.

Overall, market volatility during these times has revealed the significant influence of external macroeconomic events and geopolitical risks on the return rates of ESG sectors. This impact is reflected not only in short-term market fluctuations but also has important implications for investors’ long-term risk preferences and asset allocation decisions. Therefore, for risk managers, identifying the dynamics of interactions between ESG industries, understanding the connections within the entire financial system, and the potential pathways of risk transmission, and early identification of which industries may become ‘super-spreaders’ of risk, are crucial for protecting the stability of the entire financial system and contribute to achieving carbon neutrality goals and implementing new development concepts.

Table 2 provides descriptive statistics for various ESG industries and tests on time series data, revealing the past comprehensive performance of ESG industries. From Table 2, it can be seen that all ESG industries have shown positive average returns over the past decade, indicating that, despite experiencing the impacts of the global financial crisis or other macroeconomic fluctuations, the ESG industries have overall maintained a growth momentum. Particularly, the RT industry has exhibited the highest average returns, reflecting the stable growth in consumer spending and the continuous expansion of global trade. The PE and MED industries have also displayed higher returns, which are associated with the increasing global demand for energy efficiency and digital media content.

The RT industry not only demonstrates the highest average returns but also holds the highest ESG score. Moreover, other high-return industries like PE and MED also have very high ESG scores, which aligns with the research of Becchetti [5] and Cerqueti [6]. This indicates that these industries excel not only in financial performance but also make significant contributions to environmental protection, social responsibility, and good governance. This trend reflects a broader market phenomenon, where companies and industries that perform well in ESG aspects tend to achieve better economic performance and higher investment returns. This may be because these companies are better equipped to address global challenges, such as climate change and social inequality, making them more favored by investors.

However, despite considerable returns, the MED industry also faces the greatest risk, due to its susceptibility to technological changes and shifts in consumer habits, leading to high volatility in its returns. The NM and ELC industries also carry relatively high risks, which is related to the significant impact of global economic cycles and market supply and demand changes on the prices of their products.

### 3.2. Statistical Analysis of VaR and CoVaR

Using a Single Index Model combined with macroeconomic and microeconomic variables, VaR and CoVaR for different ESG industries were estimated and subjected to descriptive statistical analysis as shown in Table 3. These statistical results not only provide a quantitative assessment of the risk levels of ESG industries but also reveal their sensitivity to risk at both macro and micro levels.

From Table 3, it can be found that the risk value of the MED industry is significantly higher than other industries, which may be related to factors such as the media industry’s high volatility of returns, intense market competition, rapid technological changes, and constantly changing consumer preferences. The high risk value of the MED industry might reflect the instability of its business model and the uncertainty of the market environment. The risk value of the NM industry closely follows, which is related to the fluctuations of the global economic cycle, the volatility of commodity prices, and the uncertainty surrounding the demand for raw materials. Since non-ferrous metal products are widely used across various industries, their prices and demand are directly affected by the level of economic development and industrial activities, leading to increased risk volatility. The risk value of the TA industry is also relatively high, due to the industry being affected by international trade policies, changes in consumer spending, and fluctuations in the global supply chain.

From Figure 2, it can be seen that the risk value for the ESG industries in 2016 and 2020 were at relative highs within the observed time range, indicating a significant increase in market risk during these two years. This was due to the Chinese stock market crash, which led to a surge in stock market volatility, causing widespread market panic and a sharp decline in investor confidence [2]. The unwinding of high-leverage trades and a drastic shift in market sentiment exacerbated the risk contagion, not only affecting the stock market and ESG industries but also posing a threat to the stability of the entire financial system. The global pandemic of COVID-19 in 2020 caused extensive shocks to all industries. Increased uncertainty and restrictions on economic activity led to panic among market participants. Disruptions in the global supply chain and a decline in consumer demand further exacerbated market instability, thereby increasing the likelihood of risk contagion.

### 3.3. TENET Network Model Visualization Analysis

This researcher shifts the focus to the interconnectedness between ESG industries. Directional connections between industries are explored by calculating the elements $\;{\hat{D}}_{j|i}^{t}$ of the connectivity matrix. Directional connections reveal the potential influence or susceptibility of one industry on another. To more intuitively demonstrate this interconnectedness, a weighted adjacency matrix **$\hat{A}$** is used to present the results in the form of an elliptical network diagram as shown in Figure 3.

In Figure 3, the red represents the Consumption, encompassing seven different ESG industries that deal directly with consumer goods, such as FB, HA, and SS. The blue stands for Cycles, which typically have a higher correlation with the fluctuations of the economic cycle. This sector includes sixteen ESG industries, such as NM, BNK, and RE. The green signifies Technology, comprising five ESG industries covering tech giants and innovative companies, including CMP, ELC, MED, and other high-tech industries.

To more intuitively reflect the core composition of the systemic risk network and the trajectory of risk diffusion, this research considers values less than the average of the top 100 largest partial derivatives as ineffective connections and sets them to zero. This approach highlights more significant risk contagion paths and ensures that the analysis focuses on those nodes that act as key bridges within the network. The specific results are shown in Figure 4.

In Figure 4, it is clear that the impact of the Consumption and Technology sectors on the cyclical industry is the most significant. This is due to the growth of the Consumption and Technology sectors being closely linked to economic cycles, and these two sectors are also the main drivers of innovation and consumption trends. Their impact on the Cycles sector reflects the leading position of these industries in the market and their role as leading indicators in economic fluctuations.

Particularly noteworthy is the strong influence of the utilities industry on the CMP industry’s risk contagion within the entire network. This could be related to the stability of the ULT industry and the growth potential of the CMP industry, along with its sensitivity to economic fluctuations. Additionally, the risk contagion from the COAL industry to the ELC industry also shows a high impact, which is due to common risk factors in the supply chain and production processes of these two industries.

As a significant source of risk contagion, the COM industry has the most influence on the pathways of other industries, which might be because it is the infrastructure of the modern economy, and its stability is crucial to other industries. Fluctuations in the COM industry could affect the entire market through various channels, including investor confidence, production efficiency, and the adoption and application of new technologies.

Figure 5 depicts the risk contagion pathways within the Consumption, Cycles, and Technology sectors, intuitively revealing the complex structure of mutual influences between industries. This can help identify key nodes in the risk network.

Within the Consumption sector, the risk contagion connections between the TA industry and FB industry are the most numerous. This is because these industries are closely related to everyday life, and even minor shifts in consumer preferences or slight market trends can trigger rapid risk spread among them. Additionally, as a fundamental component of daily consumption, the FB industry is highly sensitive to economic fluctuations, which may lead to increased risk sensitivity.

In the Cycles sector, the COAL industry is almost at the center of the risk contagion network, reflecting its core position in the energy and raw materials supply chain. Since COAL is a basic input for many industrial processes, any fluctuations in global economic activity can transmit through the COAL industry to other industries. The RE and BM industries follow closely, which is related to their importance in the economic cycle and their close connections with other industries.

In the Technology sector, the central position of the computer industry highlights its importance in the modern economy. As a hub of technological innovation and information dissemination, the CMP industry responds quickly to market changes and has a strong risk contagion capability. Any major breakthroughs or market fluctuations in the technology sector may first impact the CMP industry and then rapidly spread to other industries such as ELC and COM.

### 3.4. Network Entropy

Figure 6 illustrates the fluctuation of network information entropy over time. By tracking the trajectory of information entropy changes, it is possible to observe the evolution of network structural complexity, as well as the dynamics of information flow within the network and interactions between nodes [59].

In 2015, influenced by the stock market crash, the significant decline in the network’s information entropy indicated an increase in the market’s predictability and a reduction in information uncertainty [2]. This might be due to market participants tending to adopt more uniform investment strategies during the stock market crash, which reduced the diversity of market behaviors. However, the increase in information entropy within the Cycles sector is because there still exists a certain degree of heterogeneity within the sector. For instance, some industries exhibit trends different from others due to specific market conditions or policy changes.

Influenced by the Sino–US trade frictions, the network’s information entropy experienced a sharp decline during this period. This decrease reflects a reduction in uncertainty and complexity within the network, as market participants reacted to the uncertainty surrounding trade policies by adopting more conservative or homogenized strategies to mitigate risk.

During the COVID-19 pandemic, the volatility of information entropy significantly increased, as countries implemented loose monetary policies, including lowering interest rates, quantitative easing, and injecting liquidity into the financial system. These measures were aimed at alleviating economic pressure, stabilizing financial markets, and promoting economic recovery. However, market participants reacted differently to the future economic conditions and policy environment, with investors and decision-makers adopting a variety of strategies to adapt to this uncertainty. Some may have become more conservative, attempting to protect their assets from potential market fluctuations, while others may have sought to seize opportunities brought about by policy-induced market changes. This diversity of strategies increased the complexity of market behavior, which was reflected in the volatility of information entropy. Therefore, the loose monetary policies during the pandemic not only had a direct impact on the economy but also exacerbated market uncertainty and complexity, thereby affecting the volatility of information entropy.

### 3.5. Time-Varying Network Structure Analysis

Figure 7 depicts the systemic risk network in-degree and out-degree for the three major sectors of industries, where red represents Consumption, blue represents Cycles, and green represents Technology. The diagram shows that within the entire system, different categorized industries exhibit synchronous fluctuations in their risk networks.

During the period from 2015 to 2016, the entire market’s volatility increased, making all ESG industries more sensitive to external risk transmission. In this period, market participants were filled with uncertainty about the future, leading to increased risk contagion and intensified risk transmission between industries. In 2018, the outbreak of the Sino–US trade frictions further exacerbated global market uncertainty. This affected not only the directly involved industries but also a broader range of sectors through supply chains and market confidence, leading to intensified risk contagion and a clear upward trend in inter-industry risk transmission. By 2020, the global pandemic of COVID-19 further escalated market uncertainty and risk levels, impacting nearly all industries. This unprecedented health crisis led to a significant slowdown in global economic activities, thereby continuously increasing risk contagion among industries [23]. By the end of 2022, as China began to lift lockdown measures, various industries started to gradually recover, market confidence was partially restored, and the risk contagion correspondingly weakened.

From Table 4 and Table 5, through in-depth analysis of the top ten ESG industries by in-degree and out-degree rankings in the systemic risk network, it can be observed that certain key industries such as COAL, CMP, COM, RT, and RE [60] play key roles in the risk contagion network. These ESG industries are not only susceptible to changes in other industries but also act as significant sources of risk contagion within the network, exerting a considerable impact on other industries. A common characteristic of these industry is their central position within their respective fields and their interactivity with a wide range of industries. Their role in the risk contagion network is akin to that of ‘hubs,’ playing a critical role in transmitting both positive and negative influences. Monitoring and analyzing these industries are crucial for understanding the overall market risk situation, predicting economic trends, and developing effective risk management strategies.

Figure 8 illustrates the trends in the relative influence (RI) of various sectors. During stock market crash, the RI of the Consumption and Technology sectors consistently remain positive, indicating that these sectors transmit more risk to the system during downturns, thus acting as net exporters of systemic risk. In contrast, the RI values for the Cycles sector is negative, suggesting that the Cycles sector is more affected by systemic risks rather than being propagators of it during economic slumps. During the COVID-19 pandemic, there was a significant shift. The RI values for the Consumption and Cycles sectors turned positive, implying that in the economic turmoil triggered by the pandemic, these sectors became net senders of systemic risk. The Consumption sector was impacted by lockdown measures and a decline in consumer confidence, while the Cycles sector was affected by disruptions in the global supply chain and a downturn in demand. Meanwhile, the RI for the Technology sector turned negative, indicating that, during the pandemic, with the surge in demand for remote work and digital services, the Technology sector became a net recipient of risk, relative to other sectors. Therefore, policymakers should adjust their strategies and decision-making accordingly to better manage potential risks and uncertainties.

Characterizing the centrality of risk contagion over time, Table 6 lists the top five ESG industries with the highest risk contagion centrality. Figure 7 depicts a time-varying heat map of the risk contagion across different industries.

The results in Table 6 show that different macroeconomic and geopolitical events can lead to various industries becoming the centers of risk contagion.

During stock market crash periods: The BNK industry is almost at the center of the entire risk contagion network. This reflects the central role of the BNK industry in the financial markets and its significant impact on market liquidity and credit risk [55]. In times of stock market crash, with increased market volatility and rising credit risks, the stability of the BNK industry is particularly crucial for the overall market.

In relatively stable periods: The FB industry occupies a central position in the entire risk contagion network. This is because during periods of economic stability, the demand for basic consumer goods is relatively stable, making the FB industry a stabilizing factor in the market.

After the Sino–US trade frictions: Technology sectors such as ELC and DMI industries become the core of the entire risk contagion network. This is due to the trade war directly affecting the market prospects of these industries, increasing the uncertainty and risk exposure between industries.

Following the outbreak of the COVID-19 pandemic: The TL and PB industries take center stage in the entire risk network. The pandemic has had profound impacts on global transportation and logistics, while also increasing the demand for pharmaceutical and biotechnology products, making these industries key nodes in the risk network.

Figure 9 displays a heat map illustrating the intensity of risk contagion between different industries and years. Between 2015 and 2016, influenced by the stock market crash, many ESG industries showed varying levels of risk contagion [61], yet not all ESG industries exhibited significant risk contagion. During 2018 to 2019, the Sino–US trade frictions had a significant impact on the Cycles sector, and during the COVID-19 pandemic, risk contagion within ESG industries was primarily concentrated in the PB and BNK industries. Throughout the entire sample period, the level of risk contagion in ESG industries remained consistent with previous analyses, demonstrating the model’s good stability.

This stability suggests that, due to the ESG industries’ focus on environmental, social, and governance factors, they may possess inherent mechanisms that help mitigate risk contagion. These mechanisms could be associated with their sustainable business practices, which might contribute to a more robust and resilient economic performance even under market pressures. Additionally, a strategic emphasis on governance may enable these industries to respond more effectively to crises, thereby limiting the spread of risks within the financial system.

### 3.6. Correlation Analysis of ESG Industry and Traditional Industry Based on the Mantel Test

To deepen our understanding of the relationship between industries that actively perform environmental, social, and corporate governance responsibilities and those that operate according to traditional practices, we will conduct Mantel tests of correlation on domestic ESG industry indices versus traditional industry indices. This testing method allows us to detect if there is a risk linkage effect between ESG industries and traditional industries. It enables us to more thoroughly understand whether the ESG indices truly reflect the sustainable development practices of businesses. Furthermore, it offers a new perspective on the actual role of sustainable development in modern corporate operations.

Figure 10 indicates that significant risk linkage effects between ESG industries and traditional industries are primarily concentrated in the RE, BNK, and CMP industries. This suggests that companies within these industries, whether ESG industries or traditional industries, may experience similar risk impacts when the market faces volatility or uncertainty. In contrast, other ESG industries appear to have stronger risk adaptation and resistance capabilities. This resilience is attributed to the proactive measures these sectors take in environmental protection, social responsibility, and governance, which help mitigate potential risks [62]. Furthermore, the risk resistance of ESG industries may also stem from sustainable resource management, deep commitment to social responsibilities, and a sensitive response to the needs of internal and external stakeholders. The performance of these sectors reinforces the value of sustainable development strategies in contemporary businesses, especially highlighting the growing importance of ESG principles when facing increasingly severe environmental challenges and social responsibility demands.

## 4. Conclusions

The TENET network model uses VaR and CoVaR to quantify tail risk, capturing not only the magnitude of tail risk but also revealing the paths and directions of risk contagion in the ESG industry. Through the TENET model, this research identifies the industries with risk concentration in the ESG industry’s risk contagion network and elucidates the mechanism of risk contagion. Moreover, it delves into the time-varying topological characteristics of the TENET network model from the perspective of information entropy.

Empirical results show the following: (1) Industries such as COAL, CMP, COM, RT, and RE play key roles in the risk contagion network. These industries are not only susceptible to changes in other industries but also act as significant sources of risk contagion within the network, exerting a considerable impact on other industries. (2) There is a dynamic interaction between the ESG industry and external economic forces, with different macroeconomic and geopolitical events causing different ESG industries to become centers of risk contagion. However, not all ESG industries exhibit significant risk contagion, and there is no apparent risk linkage effect between ESG industries and traditional industries. (3) This resilience stems from the unique business philosophies and risk management strategies of the ESG industries, which often invest more resources in environmental protection, social responsibility, and corporate governance, thus showing greater resilience in the face of market volatility and uncertainty. The robustness of these industries can positively impact investor confidence and even attract more capital inflows during market turmoil.

Due to the lack of significant volatility clustering in the ESG industries and its high-dimensional characteristics that necessitate considering all other interaction effects by incorporating more variables, this research utilizes the Single Index Model to estimate VaR and CoVaR. Furthermore, information entropy is introduced as a tool for assessing the dynamic properties of network models because it can measure the complexity and uncertainty of interactions among elements within financial networks, effectively capturing the network’s elasticity. Given the complexity and diversity of financial markets, one potential future research direction could involve selecting and applying various nonlinear models based on the specific characteristics of different markets. For example, the GARCH model can accurately capture the volatility clustering in financial markets, while the Copula model can simulate the tail dependency among assets under extreme market conditions. By integrating these models with the concept of information entropy, it is possible not only to assess the elasticity of the network at a macro level but also to more precisely quantify the risk contagion among financial institutions across different markets at a micro level. Moreover, as concepts related to information entropy, belief entropy and transfer entropy can also quantify the certainty and predictability within a system. Therefore, conducting a theoretical analysis of entropy-based network model indicators from the perspective of other entropy properties is also a worthwhile direction for exploration.

Overall, analyzing the time-varying topological characteristics of the TENET network model from the perspective of information entropy provides a novel viewpoint for understanding the mechanisms of risk contagion in financial markets. Through the entropy measurement characteristics of the TENET model, regulatory authorities can assess and monitor tail risks in financial markets more precisely, enabling them to evaluate financial stability, optimize risk management practices, and develop effective regulatory policies. 

## Figures and Tables

**Figure 1 entropy-26-00206-f001:**
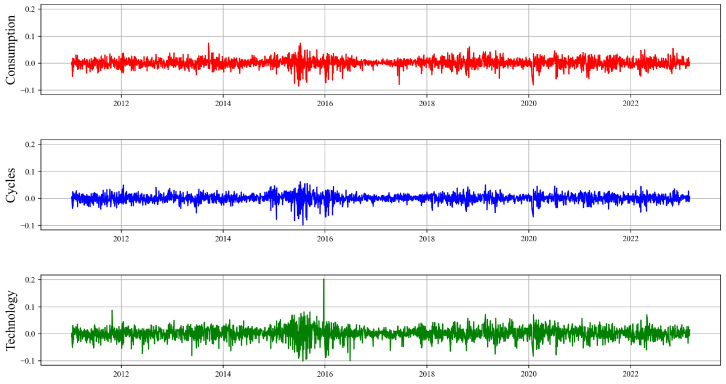
Time series graph of log returns, arranged from top to bottom, corresponds to the Consumption, Cycles, and Technology.

**Figure 2 entropy-26-00206-f002:**
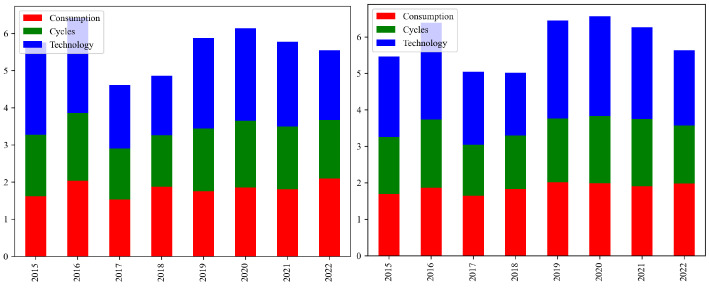
Annual average VaR and CoVaR bar chart. On the left is VaR, and on the right is CoVaR. Red represents the Consumption, blue represents the Cycles, and green represents the Technology.

**Figure 3 entropy-26-00206-f003:**
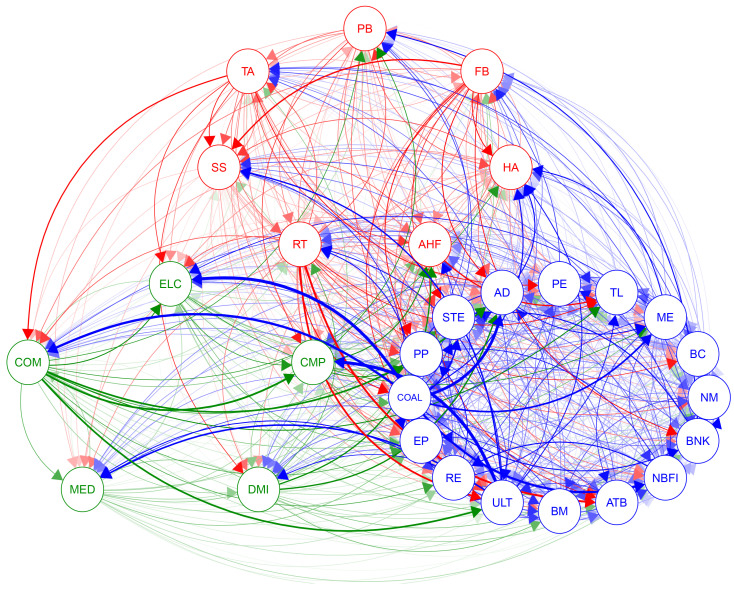
Full sample network. Red represents the Consumption, blue represents the Cycles, and green represents the Technology: α=0.05, window size n=48.

**Figure 4 entropy-26-00206-f004:**
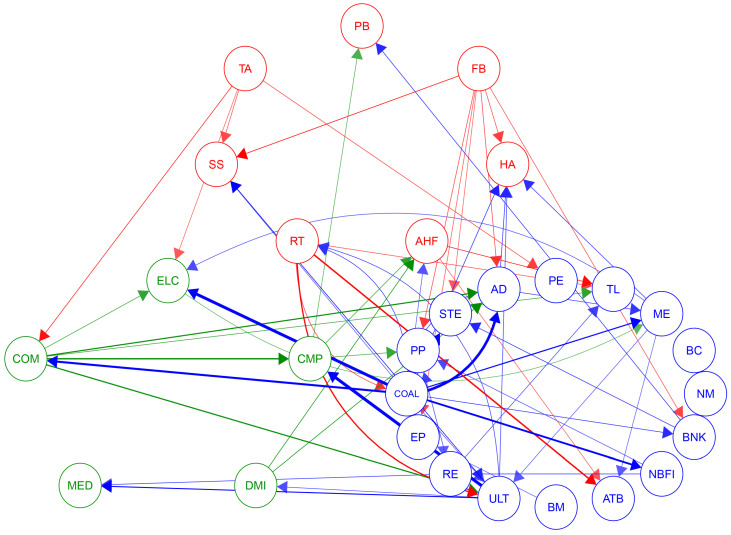
Significant sample network. Red represents the Consumption, blue represents the Cycles, and green represents the Technology: α=0.05, window size n=48.

**Figure 5 entropy-26-00206-f005:**
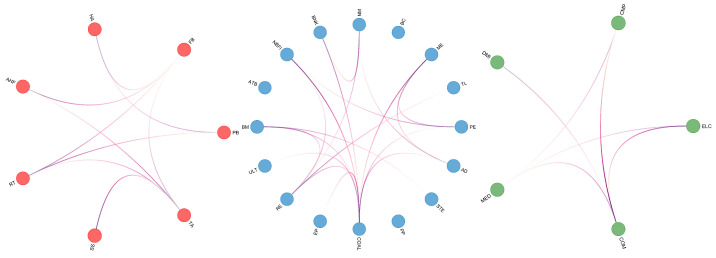
Three sectors of internal industry network, from left to right, are Consumption, Cycles, and Technology: α=0.05, window size n=48.

**Figure 6 entropy-26-00206-f006:**
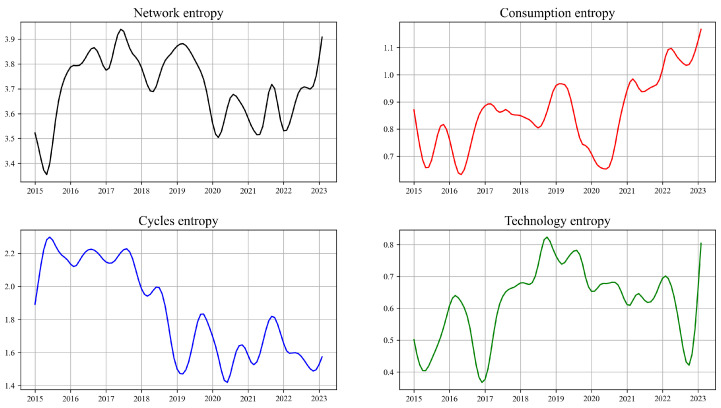
Figure 6 shows the change in information entropy over time, with four separate plots for the full Network, Consumption, Cycles, and Technology.

**Figure 7 entropy-26-00206-f007:**
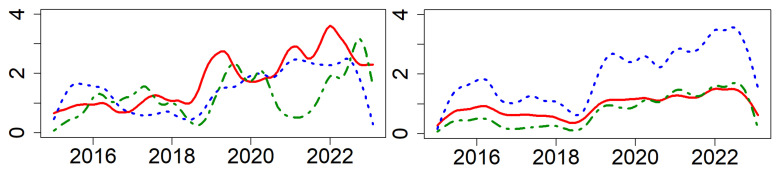
Measurement of network, where the left side represents the in-degree and the right side represents the out-degree. Red represents Consumption, blue represents Cycles, and green represents Technology: α=0.05, window size n=48.

**Figure 8 entropy-26-00206-f008:**
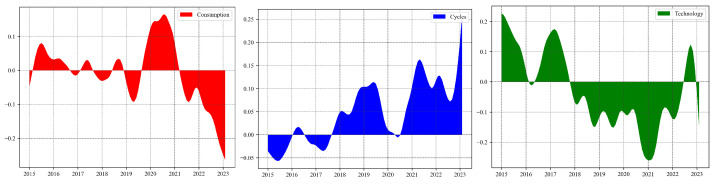
Dynamic relative influence of each sector, from left to right, is in the order of Consumption, Cycles, and Technology.

**Figure 9 entropy-26-00206-f009:**
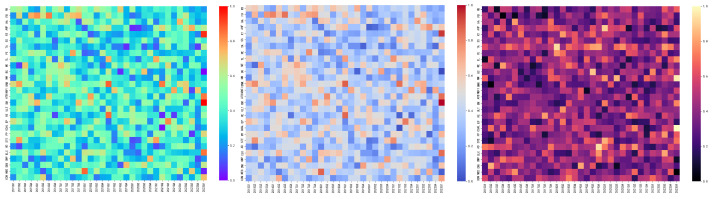
The risk contagion heat map, from left to right, is in the order of CONC, $TC^{in}$, $TC^{out}$.

**Figure 10 entropy-26-00206-f010:**
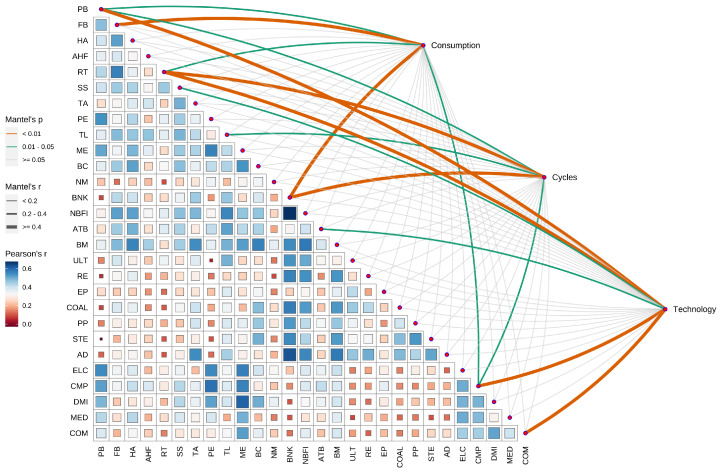
Mantel test chart for ESG and traditional industries.

**Table 1 entropy-26-00206-t001:** Industry classification table.

Sector	Industry	Short Name	Stock Number
Consumption	Pharmaceuticals and Biotechnology	PB	47
	Food and Beverage	FB	14
	Home Appliances	HA	7
	Animal husbandry and fishery	AHF	5
	Retail and Trade	RT	2
	Social Services	SS	2
	Textiles and Apparel	TA	1
Cycles	Power Equipment	PE	27
	Transportation and Logistics	TL	17
	Machinery Equipment	ME	15
	Basic Chemicals	BC	13
	Nonferrous Metals	NM	11
	Banking	BNK	11
	Non-banking Financial Institutions	NBFI	11
	Automobile	ATB	11
	Building Materials	BM	10
	Utilities	ULT	9
	Real Estate	RE	7
	Environmental Protection	EP	6
	Coal	COAL	5
	Petroleum and Petrochemical	PP	5
	Steel	STE	5
	Architectural Decoration	AD	4
Technology	Electronics	ELC	17
	Computers	CMP	10
	Defense and Military Industry	DMI	10
	Media	MED	5
	Communication	COM	4

**Table 2 entropy-26-00206-t002:** Summary statistics of daily log returns.

Industry	Mean	Std	Skew	Kurtosis	JB	ADF	ESG Score
Consumption							
PB	0.010086	0.08859	−0.13665	0.624106	633.64 ***	−15.01 ***	7.54
FB	0.015326	0.087025	−0.70071	1.733604	1797.25 ***	−42.47 ***	7.40
HA	0.006595	0.096462	−0.07841	2.431205	39,699.63 ***	−22.19 ***	6.50
AHF	0.004386	0.099432	0.096996	0.842138	17,880.20 ***	−24.10 ***	6.97
RT	0.020707	0.123142	0.574276	2.661631	173,612.12 ***	−18.97 ***	8.46
SS	0.011169	0.107611	−0.13489	2.627302	1772.35 ***	−55.93 ***	7.17
TA	0.001209	0.095757	−0.09315	4.630471	169,177.80 ***	−55.07 ***	5.95
Cycles							
PE	0.017288	0.11205	0.159149	0.624394	6589.24 ***	−54.89 ***	7.98
TL	0.005048	0.088548	0.08035	2.066187	42,861.11 ***	−26.2 ***	7.14
ME	0.003389	0.093523	−0.42513	1.83023	3414.01 ***	−42.62 ***	6.98
BC	0.012457	0.10663	−0.28994	0.30381	33,553.87 ***	−21.66 ***	7.99
NM	0.008695	0.144318	0.582843	4.77747	5153.30 ***	−55.56 ***	7.47
BNK	0.003329	0.056745	0.780311	3.665301	6805.73 ***	−10.95 ***	7.48
NBFI	0.005618	0.092892	0.022439	3.287873	37,753.71 ***	−9.45 ***	6.94
ATB	0.009188	0.080367	−0.12778	0.276506	19,123.80 ***	−13.35 ***	7.25
BM	0.003713	0.095118	−0.05981	−0.03298	4999.93 ***	−40.43 ***	7.47
ULT	0.007641	0.068062	1.004595	5.509093	55,050.59 ***	−11.69 ***	7.11
RE	0.00782	0.100706	0.251563	1.69491	1269.81 ***	−41.57 ***	6.17
EP	0.002075	0.099464	−0.2805	0.747501	23,204.43 ***	−41.23 ***	7.64
COAL	0.009856	0.082512	0.051158	0.3185	2046.22 ***	−57.39 ***	6.68
PP	0.00498	0.081159	1.39561	4.322518	10,576.62 ***	−10.46 ***	7.91
STE	0.007925	0.092402	0.408978	9.32103	44,160.83 ***	−41.55 ***	7.45
AD	0.006064	0.104453	2.088332	9.514282	6635.51 ***	−10.28 ***	7.63
Technology							
ELC	0.013052	0.125885	−0.08578	0.895578	1232.19 ***	−54.48 ***	7.65
CMP	0.009701	0.116605	0.044559	−0.12171	3907.88 ***	−17.41 ***	7.76
DMI	0.011402	0.107171	0.46677	1.22349	1766.90 ***	−54.80 ***	6.68
MED	0.017207	0.180063	1.141347	6.081793	6,117,181.22 ***	−14.34 ***	7.84
COM	0.006243	0.113371	0.419617	2.474634	19,219.10 ***	−12.60 ***	6.72

Note: (***) 1% level of significance.

**Table 3 entropy-26-00206-t003:** Summary statistics of VaR and CoVaR. The tail risk is calculated by a Single Index Model, where α=0.05 and window size n=48.

Industry	VaR Mean	VaR Std	CoVaR Mean	CoVaR Std
Consumption				
PB	−0.1313	0.0686	−0.1393	0.0371
FB	−0.1166	0.0883	−0.1191	0.0471
HA	−0.1398	0.0917	−0.1489	0.0568
AHF	−0.1491	0.0797	−0.1513	0.0456
RT	−0.1575	0.0971	−0.1766	0.0573
SS	−0.1776	0.0876	−0.1974	0.0619
TA	−0.1859	0.0982	−0.1891	0.0741
Cycles				
PE	−0.1353	0.0774	−0.1422	0.0507
TL	−0.1382	0.0807	−0.1552	0.0428
ME	−0.1528	0.0893	−0.1524	0.0679
BC	−0.1696	0.0888	−0.1777	0.0447
NM	−0.2158	0.1144	−0.2212	0.0633
BNK	−0.0881	0.0352	−0.0833	0.0222
NBFI	−0.134	0.0958	−0.1362	0.0692
ATB	−0.1042	0.0633	−0.1196	0.0297
BM	−0.1448	0.0732	−0.1458	0.0472
ULT	−0.098	0.0616	−0.0951	0.0393
RE	−0.1515	0.0852	−0.1556	0.0463
EP	−0.1617	0.0791	−0.1638	0.0409
COAL	−0.1129	0.0533	−0.1206	0.0345
PP	−0.1041	0.0467	−0.1153	0.032
STE	−0.1206	0.0653	−0.143	0.0362
AD	−0.1312	0.0673	−0.1374	0.0418
Technology				
ELC	−0.157	0.1019	−0.1853	0.0541
CMP	−0.1616	0.0775	−0.1803	0.0435
DMI	−0.1562	0.0784	−0.1624	0.0413
MED	−0.2543	0.1524	−0.2738	0.0733
COM	−0.1731	0.1108	−0.1834	0.0666

**Table 4 entropy-26-00206-t004:** Top ten ESG industries by in-degree; the received links from other industries and transmitted links to other industries are shown correspondingly. Note that only the first three most influential industries are listed.

Rank	Industry	Received Link from	Transmitted Link to	In-Degree
1	COAL	ME, CMP, BM	DMI, PE, TA	31.369
2	PE	AHF, TA, STE	COAL, RT, MED	23.160
3	DMI	RE, NM, MED	CMP, COAL, COM	22.458
4	COM	CMP, RE, TA	PE, FB, TA	22.115
5	RT	STE, BC, ATB	PE, DMI, BM	21.948
6	CMP	MED, NM, AHF	COM, PE, FB	20.745
7	BM	AHF, SS, TA	COAL, COM, PE	18.842
8	RE	AHF, NBFI, DMI	PE, DMI, BM	18.581
9	FB	COM, TA, AD	COAL, PE, RT	18.454
10	MED	ME, RT, BC	CMP, DMI, COAL	18.157

**Table 5 entropy-26-00206-t005:** Top ten ESG industries by out-degree; the received links from other industries and transmitted links to other industries are shown correspondingly. Note that only the first three most influential industries are listed.

Rank	Industry	Received Link from	Transmitted Link to	Out-Degree
1	TA	COM, PE, COAL	COAL, PE, COM	18.881
2	AD	AD, COM, STE	COAL, DMI, CMP	17.844
3	RT	STE, BC, ATB	PE, DMI, BM	17.130
4	RE	AHF, NBFI, DMI	DMI, COAL, COM	17.022
5	AHF	STE, BC, ATB	PE, CMP, RT	16.972
6	COAL	ME, CMP, BM	DMI, PE, TA	16.921
7	CMP	MED, NM, AHF	COM, PE, FB	16.608
8	NM	ULT, ME, TA	DMI, CMP, MED	16.273
9	COM	CMP, RE, TA	PE, FB, TA	16.120
10	STE	TA, MED, CMP	RT, PE, MED	15.859

**Table 6 entropy-26-00206-t006:** Top five table of risk contagion centers. The risk contagion centrality is listed based on the calculation using Equation (Equation 23).

Time	First	Second	Third	Fourth	Fifth
2015–2016	BNK	STE	PP	ATB	ELC
2016–2017	FB	BM	NM	EP	ELC
2017–2018	FB	NM	ME	BNK	PB
2018–2019	ELC	MED	PB	BM	HA
2019–2020	DMI	RE	EP	BC	PE
2020–2021	TL	EP	PB	BNK	PP
2021–2022	PB	BNK	NBFI	MED	AD
2022–2023	BM	RT	ELC	ATB	COM

## Data Availability

The data presented in this study are available on request from the corresponding author.

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
