# Peer review of "Research on Risk Contagion in ESG Industries: An Information Entropy-Based Network Approach"

_entropy, 2024, doi:10.3390/e26030206_

Round 1

Reviewer 1 Report

Comments and Suggestions for Authors

This research introduces an information entropy tail risk network model for quantifying tail risk in financial network analysis to reveal potential risk concentration points in the ESG industries. About the information entropy tail risk network model, it is relatively novel. However, I have the following questions and suggestions for this article.

1. “under the Background of Carbon Neutrality” in the title is not reflected in the study, so it is suggested to fully elaborate it in the text or delete it in the title.

2. In Abstract, the main contents of the research (including not only the research methods but also the research objects) should be pointed out.

3. In Section 1 (the introduction), I believe it will be better to supplement the research steps.

4. It is suggested that the research motivation, innovation and contribution should be summarized and highlighted in the introduction.

5. Is it necessary to combine information entropy and tail risk? I hope it can be further fully explained.

6. The transition from Formula (9) to Formula (10) is confusing, and more detailed steps will be easy for readers to understand.

7. In the first paragraph of page 10, the expression of “COVID-19 in 2021” seems unreasonable, because COVID-19 broke out at the end of 2019 and the beginning of 2020.

Author Response

We feel great thanks for your professional review work on our article. Please see the attachment.

Reviewer 2 Report

Comments and Suggestions for Authors

This paper constructs an information entropy tail risk network model, combining VaR and CoVaR as entropy metrics to quantify tail risk in financial network analysis. By analyzing the arithmetic examples, the intensity and direction of risk contagion in the ESG industry are extracted, revealing the potential risk concentration points in the ESG industry, and providing a new perspective to understand the dynamic propagation of risk within the financial network from the entropy perspective. However, the paper suffers from major shortcomings including unclear language, inappropriate content setting, and lack of innovation. The recommendations are as follows:

1. In Section 2, this paper describes that the formulae and tail risk network measurement indexes cited are not sufficiently matched to its problem-solving, resulting in the lack of rationality and scientificity of the constructed tail risk network model.

2. In Section 2.2, risk contagion is reflected in the connectivity matrix of the information entropy network as the focus of this paper. However, the setting of only taking the information entropy between financial institutions as a matrix element to consider the risk spillover effect between institutions lacks scientific description. It is suggested to add a description of the background and necessity of the relevant content.

3. This paper lacks a discussion of limitations and future research directions, and fails to explain in depth both the theoretical and practical implications of the findings. It is recommended to add a discussion to reflect the inspirational nature of this paper.

4. This paper lacks an explicit description of the practical implications and gaps, reducing the depth of exploration of this study, it is recommended be added to the Abstracts and Introduction.

5. Analysis of the existing research parts of the paper does not fully demonstrate an understanding of the relevant literature in the field, ignoring many important works on risk network published in journals such as Computers & Industrial Engineering, Engineering Applications of Artificial Intelligence, IEEE Transactions on Engineering Management, Engineering Construction and Architectural Management and so on. In addition, the current literature referenced is old and sparse, weakening the necessity and credibility of this paper. It is recommended to expand and update the literature.

Comments on the Quality of English Language

The language of this paper is logical, but some of the words used need to be carefully considered, please further embellishments.

Author Response

(The authors gave the same response as above.)

Reviewer 3 Report

Comments and Suggestions for Authors

The paper analyses, by the TENET method, the spillover risks between the various financial sectors in the Chinese context. The paper is well structured, easy to read, and offers a good contribution to the literature. Below are my suggestions for improvements:

1.  I suggest that the authors better explain the literature and better show their contribution. Therefore, I suggest including a literature section. For example, there are several papers that apply the TENET model in Chinese (Wang et al., 2018), European (Foglia and  Angelini, 2020), or other contexts (Hernande et al., 2020; Xu et al., 2021; Zhao et al., 2022).

2.  I strongly recommend better specifying which companies/sectors have higher ESG values, for example, in Table 2 as well as in the data and results section.

3. I recommend trying to link the empirical results with the literature by showing when they agree or disagree. This would make the results more solid.

4.  Only if possible: it would be interesting to construct the net RI index (see Foglia and Angelini,2020) in order to better quantify the contribution of each sector

 Reference

Foglia, M., & Angelini, E. (2020). From me to you: Measuring connectedness between Eurozone financial institutions. Research in International Business and Finance54, 101238.

Hernandez, J. A., Kang, S. H., Shahzad, S. J. H., & Yoon, S. M. (2020). Spillovers and diversification potential of bank equity returns from developed and emerging America. The North American Journal of Economics and Finance54, 101219.

Xu, Q., Zhang, Y., & Zhang, Z. (2021). Tail-risk spillovers in cryptocurrency markets. Finance Research Letters38, 101453.

Zhao, W., Fan, Y., Ji, Q., & Zhang, D. (2022). Research on financial risk spillover of the countries along the Belt and Road-Based on TENET method. Xitong Gongcheng Lilun yu Shijian/System Engineering Theory and Practice, 24-36.

Wang, G. J., Jiang, Z. Q., Lin, M., Xie, C., & Stanley, H. E. (2018). Interconnectedness and systemic risk of China's financial institutions. Emerging Markets Review35, 1-18.

Author Response

(The authors gave the same response as above.)

Round 2

Reviewer 1 Report

Comments and Suggestions for Authors

The authors have addressed all my concerns.

Author Response

We feel great thanks for your professional review work on our article.

Reviewer 2 Report

Comments and Suggestions for Authors

It's apparent that you've endeavored to revise the manuscript, and some of the concerns in my previous comments have been addressed. However, the remaining two problems have not been properly solved. The recommendations are as follows:

1. In Section 2.2, your response can reflect the scientific validity of the setting of considering inter-institutional risk spillovers using only the information entropy among financial institutions as a matrix element. However, it is recommended to summarise and directly reflect the three reasons in the paper to enhance the readability.

2. In Section 4, the first point of discussion currently added may make the scientific validity of the current paper controversial in the following way: why was Shannon entropy directly chosen as a tool for assessing the dynamic properties of a network model, when more accurate VaR measurements may be achievable with non-linear models such as the GARCH and COPULA models? It is recommended that this point be revised.

Comments on the Quality of English Language

The language is fluent, but a few words need further polishing.

Author Response

(The authors gave the same response as above.)
